# Efficient softmax approximation for GPUs

**Édouard Grave, Armand Joulin, Moustapha Cissé, David Grangier, Hervé Jégou**
Facebook AI Research
`{egrave,ajoulin,moustaphacisse,grangier,rvj}@fb.com`

## Abstract

We propose an approximate strategy to efficiently train neural network based language models over very large vocabularies. Our approach, called adaptive softmax, circumvents the linear dependency on the vocabulary size by exploiting the unbalanced word distribution to form clusters that explicitly minimize the expectation of computational complexity. Our approach further reduces the computational cost by exploiting the specificities of modern architectures and matrix-matrix vector operations, making it particularly suited for graphical processing units. Our experiments carried out on standard benchmarks, such as EuroParl and One Billion Word, show that our approach brings a large gain in efficiency over standard approximations while achieving an accuracy close to that of the full softmax. The code of our method is available at `https://github.com/facebookresearch/adaptive-softmax`.

## 1 Introduction

This paper considers strategies to learn parametric models for language modeling with very large vocabularies. This problem is key to natural language processing, with applications in machine translation (Schwenk et al., 2012; Sutskever et al., 2014; Vaswani et al., 2013) or automatic speech recognition (Graves et al., 2013; Hinton et al., 2012). In particular, Neural Network Language Models (NNLMs) have received a renewed interest in recent years, by achieving state of the art performance on standard benchmarks (Jozefowicz et al., 2016; Mikolov et al., 2010). These approaches are more costly but generalize better than traditional non-parametric models (Bahl et al., 1983; Kneser & Ney, 1995).

Statistical language models assign a probability to words given their history (Bahl et al., 1983). They are evaluated by objective criteria such as perplexity (ppl), which directly measures the ability of the system to determine proper probabilities for all the words. This potentially makes parametric models prohibitively slow to train on corpara with very large vocabulary. For instance, the vocabulary of the One Billion Word benchmark (Chelba et al., 2013) contains around 800K words. In standard NNLMs, such as feedforward networks (Bengio et al., 2003a) or recurrent networks (Mikolov et al., 2010), computing this probability over the whole vocabulary is the bottleneck. Many solutions have been proposed to reduce the complexity of this expensive step (Bengio et al., 2003b; Goodman, 2001a; Gutmann & Hyvärinen, 2010). We distinguish (i) the methods that consider the original distribution and aim at providing approximations of the probabilities, or of a subset of them (Bengio et al., 2003b; Ji et al., 2015), from (ii) the approaches that compute exact probabilities for an approximate model yielding a lower computational cost, such as the popular hierarchical softmax (Goodman, 2001a; Mnih & Hinton, 2009; Morin & Bengio, 2005).

Our approach, called adaptive softmax, belongs to the second category. More specifically, it is inspired by the hierarchical softmax and its subsequent variants. In contrast to previous works and motivated by the trend that GPUs are comparatively more and more performant than CPUs, our design is oriented towards efficient processing on GPUs. In this context, our paper makes the following points:

- We define a strategy to produce an approximate hierarchical model. It departs from previous ones in that it explicitly takes into account the complexity of matrix-matrix multiplications on modern architectures, which is not trivially linear in the dimensions of the matrices.

- We conduct an empirical complexity analysis of this model on recent GPUs. This leads us to define a realistic complexity model that is incorporated in the proposed optimization;

- Our approach provides a significant acceleration factor compared to the regular softmax, i.e., $2\times$ to $10\times$ speed-ups. Equivalently we improve the accuracy under computational constraints. Importantly, on the largest corpus, this higher efficiency empirically comes at no cost in accuracy for a given amount of training data, in contrast to concurrent approaches improving the efficiency.

This paper is organized as follows. Section 2 briefly reviews the related work and Section 3 provides some background on the language modeling task that we consider. Section 4 describes our proposal, which is subsequently evaluated in Section 5 on typical benchmarks of the language modeling literature, including Text8, Europarl and One Billion Word datasets.

## 2 RELATED WORK

Many methods have been proposed to approximate the softmax efficiently (Bengio et al., 2003b; Goodman, 2001a; Gutmann & Hyvärinen, 2010; Morin & Bengio, 2005). We briefly describe the most popular ones below and point the reader to Chen et al. (2015) for a comparative study. For the sake of completeness, we refer the reader to other strategies that can speed-up the training of language models in complementary manners (Mikolov et al., 2011b).

**Loss function approximation.**  The *Hierarchical Softmax* (HSM) is an approximation of the softmax function introduced by Goodman (2001a). This approach is generally used with a two-level tree (Goodman, 2001a; Mikolov et al., 2011c) but has also been extended to deeper hierarchies (Morin & Bengio, 2005; Mnih & Hinton, 2009). In general, the hierarchy structure is built on word similarities (Brown et al., 1992; Le et al., 2011; Mikolov et al., 2013) or frequency binning (Mikolov et al., 2011c). In particular, Mikolov et al. (2013) proposes an optimal hierarchy by constructing a Huffman coding based on frequency. However this coding scheme does not take into account the theoretical complexity reduction offered by matrix-matrix multiplication and distributed computation, in particular with modern GPUs.

Similar to our work, Zweig & Makarychev (2013) constructs their hierachy in order to explicitly reduce the computational complexity. They also solve the assignment problem with dynamic programming. However, they only consider hierachies where words are kept in the leaves of the tree, leading to a significant drop of performance (reported to be around $5 - 10\%$), forcing them to also optimize for word similarity. In our case, allowing classes to be stored in the internal node of the tree leads to almost no drop of performance. Also, they assume a linear cost for the vector-matrix operation which significantly limits the use of their approach on distributed system such as GPU.

The idea of keeping a short-list of the most frequent words has been explored before (Le et al., 2011; Schwenk, 2007). In particular, Le et al. (2011) combines a short-list with a hierachical softmax based on word representation. In contrast, the word hierarchy that we introduce in Section 4 explicitly aims at reducing the complexity.

Our work also shares similarities with the *d-softmax* introduced by Chen et al. (2015). They assign capacity to words according to their frequency to speed up the training. Less frequent words have smaller classifiers than frequent ones. Unlike our method, their formulation requires accessing the whole vocabulary to evaluate the probability of a word.

**Sampling based approximation.**  Sampling based approaches have been successfully applied to approximate the softmax function over large dictionaries in different domains, such as language modeling (Jozefowicz et al., 2016), machine translation (Jean et al., 2015) and computer vision (Joulin et al., 2015). In particular, importance sampling (Bengio & Senécal, 2008; Bengio et al., 2003b) selects a subset of negative targets to approximate the softmax normalization. Different schemes have been proposed for sampling, such as the unigram and bigram distribution (Bengio et al., 2003b) or more recently, a power-raised distribution of the unigram (Ji et al., 2015; Mikolov et al., 2013). While this approach often leads to significant speed-up at train time, it still requires to evaluate the full softmax at test time.

**Self-normalized approaches.** Self-normalized approaches aim at learning naturally normalized classifier, to avoid computing the softmax normalization. Popular methods are Noise Contrastive Estimation (Gutmann & Hyvärinen, 2010; Mnih & Teh, 2012; Vaswani et al., 2013) or a penalization on the normalization function (Andreas & Klein, 2014; Devlin et al., 2014). Noise Contrastive Estimation (Gutmann & Hyvärinen, 2010) replaces the softmax by a binary classifier distinguishing the original distribution form a noisy one. While the original formulation still requires to compute the softmax normalization, Mnih & Teh (2012) shows that good performance can be achieved even without it.

Finally, Vincent et al. (2015) have also proposed an efficient way to train model with high dimensional output space. Their approach is exact and leads to a promising speed-up but it cannot be directly applied to the softmax function, limiting its potential application to language modeling.

# 3 PRELIMINARIES ON LANGUAGE MODELING

The goal of language modeling is to learn a probability distribution over a sequence of words from a given dictionary $\mathcal{V}$. The joint distribution is defined as a product of conditional distribution of tokens given their past (Bahl et al., 1983). More precisely, the probability of a sequence of $T$ words $w_1, \ldots, w_T \in \mathcal{V}^T$ is given as

$$P(w_1, \ldots, w_T) = \prod_{t=1}^{T} P(w_t \mid w_{t-1}, \ldots, w_1). \tag{1}$$

This problem is traditionally addressed with non-parameteric models based on counting statistics (Goodman, 2001b). In particular, smoothed N-gram models (Bahl et al., 1983; Katz, 1987; Kneser & Ney, 1995) achieve good performance in practice (Mikolov et al., 2011a), especially when they are associated with cache models (Kuhn & De Mori, 1990). More recently, parametric models based on neural networks have gained popularity for language modeling (Bengio et al., 2003a; Jozefowicz et al., 2016; Mikolov et al., 2010). They are mostly either feedforward networks (Bengio et al., 2003a) or recurrent networks (Mikolov et al., 2010).

**Feedforward network.** In a standard feedforward network for language modeling, we fix a window of length $N$ and predict the next words according to the words appearing in this window. In the simplest case, this probability is represented by a 2-layer neural network acting on an input $x_t \in \mathcal{V}^N$, defined as the concatenation of the one-hot representation of the $N$ previous words, $w_{t-N+1}, \ldots, w_t$. The state $h_t$ of the hidden layer and subsequently the vector of scores $y_t$ associated with the next token $w_{t+1}$ are computed as

$$h_t = \sigma(APx_t), \tag{2}$$
$$y_t = f(Bh_t), \tag{3}$$

where $\sigma$ is a non linearity, e.g., the pointwise sigmoid function $\sigma(z) = 1/(1 + \exp(-z))$, and $f$ is the softmax function discussed in the next section. This model is parameterized by the weight matrices $P$, $A$ and $B$ and is routinely learned with an optimization scheme such as stochastic gradient descent or Adagrad (Duchi et al., 2011).

**Recurrent network.** A Recurrent network (Elman, 1990) extends a feedforward network in that the current state of the hidden layer also depends on its previous state. The hidden state $h_t$ is updated according to the equation $h_t = \sigma(Aw_t + Rh_{t-1})$, where $R$ is a weight matrix and $x_t$ is the one-hot representation of the current word $w_t$. Computing the exact gradient for this model is challenging but it is possible to compute an efficient and stable approximation of it, using a truncated back-propagation through time (Werbos, 1990; Williams & Peng, 1990) and norm clipping (Mikolov et al., 2010).

Since the model introduced by Elman (1990), many extensions have been proposed, such as Longer Short Term Memory (LSTM) (Hochreiter & Schmidhuber, 1997), Gated recurrent units (Chung et al., 2014) or structurally constrained network (Mikolov et al., 2014). These models have been successfully used in the context of language modeling (Jozefowicz et al., 2016; Mikolov et al., 2010; Mikolov & Zweig, 2012). In this work, we focus on the standard word level LSTM architecture since it has obtained state of the art performance on the challenging One Billion Word Benchmark (Jozefowicz et al., 2016).

**Class-based hierarchical softmax.** In neural language modeling, predicting the probability of the next word requires to compute scores for every word in the vocabulary and to normalize them to form a probability distribution. This is typically achieved by applying a softmax function to the unnormalized score $z_w$ associate with each word $w$, where the softmax function is defined as

$$f(z_w) = \frac{\exp(z_w)}{\sum_{w' \in \mathcal{V}} \exp(z_{w'})}. \tag{4}$$

For a vocabulary comprising $k = |\mathcal{V}|$ words, this function requires $\mathcal{O}(k)$ operations once the scores are computed. In the case of neural networks, the overall complexity is $\mathcal{O}(dk)$, where $d$ is the size of the last hidden layer. When the vocabulary is large, this step is computationally expensive and often dominates the computation of the whole model (Jozefowicz et al., 2016; Mikolov et al., 2014), as discussed in introduction and related work. A simple approach (Goodman, 2001a) to reduce this computational cost is to assign each word $w$ of the vocabulary to a unique class $\mathcal{C}(w)$ and to factorize the probability distribution over words as

$$p(w_t \mid h_t) = p_1(\mathcal{C}(w_t) \mid h_t) \times p_2(w_t \mid \mathcal{C}(w_t),\ h_t),$$

where $p_1$ and $p_2$ are obtained using the softmax function (Eq. 4). If each class contains $\sqrt{k}$ words, the computational cost is reduced from $\mathcal{O}(dk)$ to $\mathcal{O}(d\sqrt{k})$.

# 4 OUR APPROACH: THE ADAPTIVE SOFTMAX

In this section, we propose the adaptive softmax, a simple speedup technique for the computation of probability distributions over words. The adaptive softmax is inspired by the class-based hierarchical softmax, where the word classes are built to minimize the computational complexity. Our method is designed to be efficient for GPUs, which are commonly used to train neural networks. For the sake of clarity, we first present the intuition behind our method in the simple case where we simply split our dictionary in two distinct clusters, before analyzing a more general case.

## 4.1 GPU COMPUTATIONAL MODEL

The bottleneck of the model described in the previous section is the matrix multiplication between the matrix representing the hidden states (of size $B \times d$, where $B$ denotes the batch size), and the matrix of word representations, of size $d \times k$. For a fixed size $d$ of the hidden layer, we denote by $g(k, B, d)$ the complexity of this multiplication, and simplify the notation wherever some parameters are fixed. Figure 1 reports empirical timings as a function of $k$ for typical parameters of $B$ and $d$ for two GPU models, namely K40 and Maxwell. We observe that the complexity $g(k)$ is constant for low values of $k$, until a certain inflection point $k_0 \approx 50$, and then becomes affine for values $k > k_0$. This suggests a cost function of the form

$$g(k) = \max(c + \lambda k_0, c + \lambda k) = c_{\mathrm{m}} + \max\left[0, \lambda(k - k_0)\right]. \tag{5}$$

Empirically, $c_{\mathrm{m}} = 0.40\,\mathrm{ms}$ on a K40 and $0.22\,\mathrm{ms}$ on a Maxwell. We observe the same behavior when measuring the timings as a function of the other parameters, *i.e.*, it is inefficient to matrix-multiply when one of the dimensions is small. This observation suggests that hierarchical organizations of words with a low number of children per node, such as binary Huffman codes, are highly suboptimal.

## 4.2 INTUITION: THE TWO-CLUSTERS CASE

In natural languages, the distribution of the words notoriously follows a Zipf law (Zipf, 1949). Most of the probability mass is covered by a small fraction of the dictionary, *e.g.*, $87\%$ of the document is covered by only $20\%$ of the vocabulary in the Penn TreeBank. Similar to the frequency binning hierarchical softmax (Mikolov et al., 2011c), this information can be exploited to reduce the computation cost.

A simple strategy to reduce the overall complexity is to partition the dictionary $\mathcal{V}$ into two clusters as $\mathcal{V}_{\mathrm{h}}$ and $\mathcal{V}_{\mathrm{t}}$, where $\mathcal{V}_{\mathrm{h}}$ denotes the *head* of the distribution consisting of the most frequent words, and where $\mathcal{V}_{\mathrm{t}}$ is the *tail* associated with a large number of rare words. The classifier frequently accesses the head, which motivates the fact that it should be computed efficiently. In contrast, the tail occurs

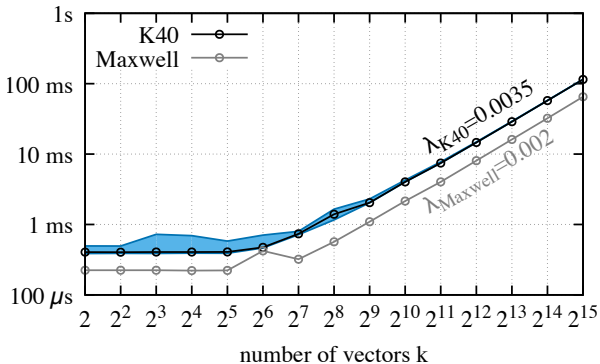

Figure 1: GPU timings for multiplying two matrices in the dominant step of the RNN model. We consider matrices of size $2560 \times 2048$ and $2048 \times k$ representing hidden states and word representations. We report the timings as a function of $k$ (number of word representations), and we compute the averages (circles) over $1000$ measures, and the minima and maxima for the K40. The standard deviation does not exceed $5\%$ of each timing.

less frequently and the corresponding computation can be slower. This suggests to define clusters with unbalanced cardinalities $|\mathcal{V}_h| \ll |\mathcal{V}_t|$ and probabilities $P(\mathcal{V}_h) \gg P(\mathcal{V}_t)$, where $P(\mathcal{A}) = \sum_{w \in \mathcal{A}} p_i$ is the probability of a word to occur in the set $\mathcal{V}_i$. For instance, one may define the head would only contain $20\%$ of the vocabulary (covering for $87\%$ on PennTree Bank). These two clusters can be organized in two different ways: either they are both leaves of a 2-level tree (Mikolov et al., 2011c), or the head cluster is kept as a *short-list* in the root node (Le et al., 2011). We now analyze what is the best structure and how to split the vocabulary by determining the corresponding complexities, assuming that the head consists of the most frequent words. The next subsection shows the optimality of this choice.

Given a vocabulary of $k$ words, we are looking for the number $k_h = |\mathcal{V}_h|$ of words from the head of the distribution to be assigned to the first cluster. These words will cover for $p_h$ of the distribution. The tail cluster will then contain the rest of the vocabulary, made of $k_t = k - k_h$ words and covering for $p_t = 1 - p_h$ of the overall distribution. We denote by $g(k, d)$ the computational complexity of computing the softmax function over $k$ words with $d$ dimensional input features. Figure 1 shows an example of this function for a fixed $d$. The complexity of putting the head of the distribution in the root of the tree is $g(k_h + 1, d) + p_t g(k_t, d)$, while the complexity associated with putting both cluster in leaves is $g(2, d) + p_h g(k_h, d) + p_t g(k_t, d)$. Depending on the distribution of a corpus, it is then simple to choose the best assignment of words into the two clusters. For example, on PennTree Bank, with a hidden layer of size $d = 128$, the optimal configuration is to keep a short-list of $1400$ classes in the root node, leading to an average cost of $0.33$ ms per batch of size $512$, while it takes $0.36$ ms when both clusters are in the leaves. In comparison, the full softmax takes $0.80$ ms for the same configuration, leading to a $\times 2.4$ speed-up.

**Adapting the classifier capacity for each cluster.** Each cluster is accessed independently of each other, they thus do not need to have the same capacity. Frequent words need high capacity to be predicted correctly. In contrast, rare words cannot be learned very well, since we only see them a few times. It would then be wasteful to associate them with high capacity. Like in Chen et al. (2015), we exploit this observation to further reduce the computational cost of our classifier. Unlike Chen et al. (2015), we share the state of hidden layer across clusters and simply reduce the input size of the classifiers by applying a projection matrix. Typically, the projection matrix for the tail cluster reduces the size from $d$ to $d_t = d/4$, reducing the complexity from $g(k_t, d)$ to $g(d_t, d) + g(k_t, d_t)$.

**Compromising between efficiency and accuracy.** We observe empirically that putting all the clusters in the leaves of the tree leads to a significant drop of performance (around $5 - 10\%$ performance drop, Mikolov et al., 2011c; Zweig & Makarychev, 2013). The reason is that the probability of every word $w$ belonging to a cluster $c$ is multiplied by the probability of its class, i.e., it is equal to $P(c \mid h)P(w \mid c, h)$, while attaching a frequent word directly to the root associates it directly to the probability $P(w \mid h)$ making its inference sharper. For this reason, unless there is a significant difference in computational complexity, we favor using a short-list, over the standard 2-level hierarchical softmax.

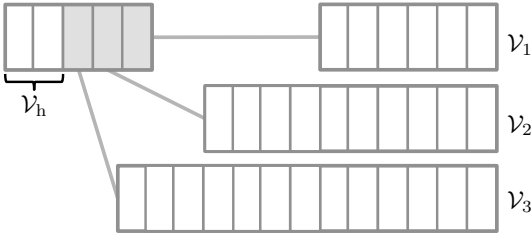

Figure 2: Our hierarchical model is organized as (i) a first level that includes both the most frequent words and vectors representing clusters, and (ii) clusters on the second level that are associated with rare words, the largest ones being associated with the less frequent words. The sizes are determined so as to minimize our cost model on GPU.

## 4.3 GENERAL CASE

Let us now consider the more general case where the dictionary is partitioned as $\mathcal{V} = \mathcal{V}_\mathrm{h} \cup \mathcal{V}_1 \ldots \mathcal{V}_J$, $\mathcal{V}_i \cap \mathcal{V}_j = \emptyset$ if $i \neq j$. We consider the hierarchical model depicted in Figure 2, where the sub-dictionary $\mathcal{V}_\mathrm{h}$ is accessed at the first level, and the others in the second level. We now investigate the computational cost $C$ of the forward (equivalently, backward) pass of this approximate softmax layer. For the time being, we fix the batch size $B$ and the dimensionality $d$ of the hidden layer, in order to analyze the complexity as a function of the sub-dictionary sizes and probabilities. We denote by $p_i = \sum_{w \in \mathcal{V}_i} p(w)$ the probability $P(w \in \mathcal{V}_i)$ and $k_i = |\mathcal{V}_i|$ the cardinality of each cluster.

The expected cost $C$ is decomposed as $C = C_\mathrm{h} + \sum_i C_i$, where

$$C_\mathrm{h} = p_\mathrm{h}\, g(J + k_\mathrm{h}) \text{ and } \forall i,\ C_i = p_i \big[ g(J + k_\mathrm{h}) + g(k_i) \big], \tag{6}$$

leading to

$$C = g(J + k_\mathrm{h}) + \sum_i p_i\, g(k_i). \tag{7}$$

We add the constraint $k \geq k_0$ to ensure that there is no penalty induced by the constant part of the cost model of Equation 5, the previous equation simplifies as

$$C = c + \lambda(J + k_\mathrm{h}) + \sum_i p_i(c + \lambda k_i) \tag{8}$$

$$= c(2 - p_\mathrm{h}) + \lambda \big[ J + k_\mathrm{h} + \sum_i p_i\, k_i \big]. \tag{9}$$

Let us discuss this equation, by first considering that the cardinalities of the sub-vocabularies are fixed. The right-most term is the only one that depends on the word probabilities. For two distinct clusters $\mathcal{V}_i$ and $\mathcal{V}_j$, we can re-write $p_j k_j$ as $(p_{i+j} - p_i)k_j$, where $p_{i+j} = p_i + p_j$, so that

$$p_i k_i + p_j k_j = p_i(k_i - k_j) + p_{i+j} k_j. \tag{10}$$

Without loss of generality, we assume that $k_i > k_j$. The quantities $p_{i+j}$, $k_i$ and $k_j$ being fixed, the second term of the right-hand side of this equation is constant, and the best strategy is trivially to minimize the probability of the largest cluster $\mathcal{V}_i$. In other terms, an optimal solution for Equation 9 requires that the most frequent words are assigned to the smallest cluster. This remark is true for any tuple $(i, j)$, and we easily see that this point also holds for the head cluster. As a consequence, for a fixed number of clusters of given sizes, the best strategy is to assign the words by decreasing probabilities to clusters of increasing size. Note, this analysis remains valid as long as the $g$ is monotonically increasing in $k$.

**Determining $k_i$ with $J$ fixed: dynamic programming.** We now assume that the number of clusters is fixed. Following our analysis above, the optimization solely depends on the cardinalities $k_i$ for all clusters, which perfectly determines how to split the list of words ordered by frequency. We solve this problem by dynamic programming.

**Finding the number of clusters.** The only remaining free variable in our optimization is $J$, since the other parameters are then determined by the aforementioned optimizations. For this step, the cost of Equation 9 over-estimates the number of clusters because we have neglected the effect of the non-linearity of the batch size: in the second layer, the batches are typically smaller than the inflection

|                | ppl | training time |
|----------------|-----|---------------|
| full softmax   | 144 | 83 min        |
| sampling       | 166 | 41 min        |
| HSM (freq)     | 166 | 34 min        |
| D-softmax      | 195 | 53 min        |
| D-softmax [*]  | 147 | 54 min        |
| **Ours**       | 147 | 30 min        |

Table 1: Text8. Perplexity and training time after 5 epochs. Our approach is significantly better than other published approximate strategies. We also show that improving the baseline D-softmax [*] as discussed in text improve the results, but is slower than our proposal.

Note, approximate strategies are comparatively less interesting for small vocabularies such as in this case.

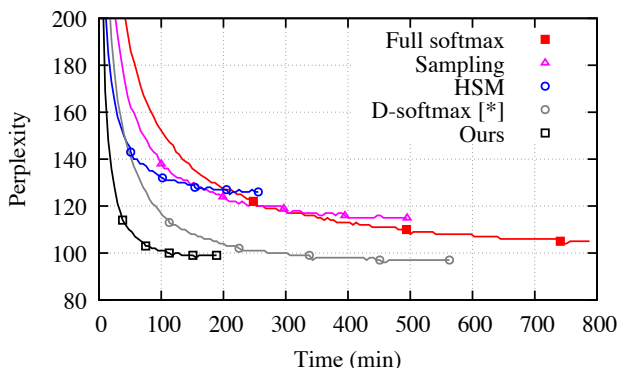

Figure 3: Training on Europarl (Finnish): perplexity (on validation) as the function of time for our method and approaches from the state of the art. We represent the result after each epoch by a point. Our method favorably compares with all other approaches w.r.t. the tradeoff perplexity and training time, and of training data *vs* perplexity. Similar conclusions are drawn for the other languages.

point $k_0$. In practice, we optimize over small values of $J = 1, 2, 3, 4$ and empirically determine the best compromise speed/perplexity on training data. Note, having a lower number of clusters with numerous frequent words on the first level has another flavor: we empirically observe that it offers a better perplexity than word hierarchy with a large number of clusters. It is comparable to that of the exact softmax for large corpora, as shown later by our experiments.

## 5 EXPERIMENTS

This section provides a set of experiments aiming at analyzing the trade-off between actual complexity and effectiveness of several strategies, in particular the approach presented in the previous section. First we describe our evaluation protocol, then we evaluate some of the properties of our model and finally we compare it on standard benchmark against standard baselines.

**Datasets.** We evaluate our method on standard datasets, and use the perplexity (ppl) as an evaluation metric, as the function of the training time or of the number of training data (epochs). The datasets have varying vocabulary sizes, in different languages, which allows us to better understand the strengths and weaknesses of the different approaches.

- Text8[1] is a standard compression dataset containing a pre-processed version of the first 100 million characters from Wikipedia in English. It has been recently used for language modeling (Mikolov et al., 2014) and has a vocabulary of 44k words.
- Europarl[2] is a machine translation corpus, containing 20 languages (Koehn, 2005). For most languages, there are 10M–60M tokens and the vocabulary is in between 44k and 250k words.
- One Billion Word [3] is a massive corpus introduced by Chelba et al. (2013). It contains 0.8B tokens and a vocabulary comprising almost 800k words.

**Implementation details.** We use an LSTM with one layer in all our experiments. On Text8 and Europarl, the models have $d = 512$ hidden units and are regularized with weight decay ($\lambda = 10^{-6}$).

---

[1]http://mattmahoney.net/dc/textdata
[2]http://www.statmt.org/europarl/
[3]https://code.google.com/archive/p/1-billion-word-language-modeling-benchmark/

| Model | Test perplexity |
|---|---|
| Interpolated Kneser-Ney 5-gram (Chelba et al., 2013) | 67.6 |
| Feedforward NN + D-Softmax (Chen et al., 2015) | 91.2 |
| 4-layer IRNN-512 (Le et al., 2015) | 69.4 |
| RNN-2048 + BlackOut sampling (Ji et al., 2015) | 68.3 |
| Sparse Non-negative Matrix Language Model (Shazeer et al., 2015) | 52.9 |
| RNN-1024 + MaxEnt 9-gram (Chelba et al., 2013) | 51.3 |
| LSTM-2048-512 (Jozefowicz et al., 2016) | 43.7 |
| 2-layer LSTM-8192-1024 + CNN inputs (Jozefowicz et al., 2016) | 30.0 |
| **Ours** (LSTM-2048) | 43.9 |
| **Ours** (2-layer LSTM-2048) | 39.8 |

Table 2: One Billion Word benchmark. Perplexity on the test set for single models. Our result is obtained after 5 epochs.

On the One Billion Word benchmark, we use $d = 2048$ hidden units and no regularization. The dimension of the input word embeddings is set to 256, so that large models fit in GPU memory. For the backpropagation through time, we unroll the models for 20 steps. We use Adagrad (Duchi et al., 2011), with a step size of 0.1 and 5 epochs, and we clip the norm of the gradients to 1. The batch size $B$ is set to 128, except on the Finnish portion of Europarl where $B$=64 due to memory constraints. All the experiments were run on the same GPU with the Maxwell architecture.

**Baselines.** Our method is compared to: (1) the full softmax, (2) the hierarchical softmax (HSM) with frequency binning (Mikolov et al., 2011b), (3) importance sampling (Bengio et al., 2003b; Bengio & Senécal, 2008) and (4) the differentiated softmax (Chen et al., 2015). For HSM, we tried different strategies for the binning. We observe that using the square root function on the count before computing the word bins is the most efficient. For the negative sampling method, we used a number of samples equal to 20% of the size of the vocabulary (Chen et al., 2015). For the differentiated softmax (D-softmax), we used the same partitions for the vocabulary as for our approach. We tried two version of the differentiated softmax. The first is the one described by Chen et al. (2015), where each word cluster uses a disjoint subset of the hidden representation. We also present an improved version, referred to as D-softmax [*], which uses our choice to have the whole hidden representation mapped to the different word clusters using projection matrices of different sizes.

**Comparison with the state of the art.** Table 1 reports the results that we achieve on Text8. On this small vocabulary, approximate methods are comparatively less interesting. Our approach is the only one to approach the result of the full soft-max (below by 3 points of perplexity), while being the fastest. Our improved variant D-softmax [*] of the work by Chen et al. (2015) obtains similar results but is slower by a factor $\times 1.8$.

On Europarl, we first present the convergence properties of our approach compared to other approximate strategies in Figure 3 show the perplexity (ppl) as a function of training time. Our approach significantly outperforms all competitors by a large margin. For reference, we also show the performance (D-softmax [*]) obtained by improving the D-softmax, to make it more comparable to our method. Our method is $2\times$ to $3\times$ faster than this improved competitor, which demonstrates how critical is our optimization strategy. Similar conclusions are drawn from Table 3 for other languages from the Europal corpus.

Table 2 gives the test perplexity on One Billion Word benchmark: Our method achieves a perplexity of 43.9 after five epochs, taking less than three days to train on a single GPU. In comparison, only Jozefowicz et al. (2016) achieves a lower perplexity, but with a model $8\times$ bigger than ours and trained over 32 GPUs during 3 weeks. We also note that for models of similar size, we achieve similar perplexity than the method introduced by Jozefowicz et al. (2016). As far as we know, ours the first method to achieve a perplexity lower than 50 on a single GPU.

| Language: | bg | | cs | | da | | de | | el | | es | |
|---|---|---|---|---|---|---|---|---|---|---|---|---|
| $k=$ | 50k | | 83k | | 128k | | 143k | | 100k | | 87k | |
| Method | ppl | $t$ | ppl | $t$ | ppl | $t$ | ppl | $t$ | ppl | $t$ | ppl | $t$ |
| Full | 37 | 58 | 62 | 132 | 37 | 713 | 42 | 802 | 38 | 383 | 30 | 536 |
| Sampling | 40 | 29 | 70 | 53 | 40 | 247 | 45 | 262 | 41 | 144 | 32 | 217 |
| HSM (freq) | 43 | 17 | 78 | 29 | 42 | 114 | 51 | 124 | 45 | 73 | 34 | 110 |
| D-softmax | 47 | 36 | 82 | 75 | 46 | 369 | 56 | 397 | 50 | 211 | 38 | 296 |
| D-softmax [*] | 37 | 36 | 62 | 76 | 36 | 366 | 41 | 398 | 37 | 213 | 29 | 303 |
| **Ours** | 37 | 18 | 62 | 30 | 35 | 105 | 40 | 110 | 36 | 72 | 29 | 103 |

Table 3: Europarl. Perplexity after 5 epochs for different languages as a function of time $t$ (minutes).

## 6 CONCLUSION

In this paper, we have proposed a simple yet efficient approximation of the softmax classifier. To our knowledge, it is the first speed optimizing approximation that obtains performance on par with the exact model. This is achieved by explicitly taking into account the computational complexity of parallel systems and combining it with a few important observations, namely keeping a short-list of frequent words in the root node (Schwenk, 2007) and reducing the capacity of rare words (Chen et al., 2015). In all our experiments on GPU, our method consistently maintains a low perplexity while enjoying a speed-up going from $2\times$ to $10\times$ compared to the exact model. This type of speed-up allows to deal with extremely large corpora in reasonable time and without the need of a large number of GPUs. We believe our approach to be general enough to be applied to other parallel computing architectures and other losses, as well as to other domains where the distributions of the class are unbalanced.

ACKNOWLEDGMENTS

The authors would like to thank Jeff Johnson for his help with GPU benchmarking and Tomas Mikolov for insightful discussions.

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
