# Peer review of "Efficient Softmax Approximation for GPUs"

_ICLR 2017 — rejected_

[Reviewer Comment · AnonReviewer2 · 01 Dec 2016]
**Notation**

g() is used in a number of different configurations with different variables. Please provide corresponding definitions (e.g. g(k,B,d) vs. g(k) same apart from keeping B and d constant in g(k)?).

[Reviewer Comment · AnonReviewer2 · 01 Dec 2016]
**Missing references/citations**

In many places, references/citations seem to be erroneous/missing (cf. "(?)" in the text). Please provide these.

[Official Review · AnonReviewer2 · rating 7 · confidence 4 · 11 Dec 2016]

he authors provide an interesting, computational-complexity-driven approach for efficient softmax computation for language modeling based on GPUs. An adaptive softmax approach is proposed based on a hierarchical model. Dynamic programming is applied to optimize the structure of the hierarchical approach chosen here w.r.t. computational complexity based on GPUs. 

However, it remains unclear, how robust the specific configuration obtained from dynamic programming is w.r.t. performance/perplexity. Corresponding comparative results with perplexity-based clustering would be desirable. Especially, in Sec. 5, Paragraph Baselines, and Table 1, respectively, it would be interesting to see a result on HSM(PPL) (cf. Zweig et al. 2013).

AFAIK, the first successful application of an LSTM-based language model for large vocabulary was published by Sundermeyer et al. 2012 (see below), which is missing in the sumary of prior work on the bottom of p. 3.

Mainly, the paper is well written and accessible, though notation in some cases should be improved, see detailed comments below.

Prior work on LSTM language modeling: 
 - Sundermeyer et al.: LSTM Neural Networks for Language Modeling, Interspeech, pp. 194-197, 2012.

Notation:
 - use of g(k) vs. g(k,B,d): g(k) should be clearly defined (constant B and d?)
 - notation should not be reused (B is matrix in Eq. (3), and batch size in Sec. 4.1).
 - notation p_{i+j} (Eq. (10) and before) is kind of misleading, as p_{i+j} is not the same as p_{(i+j)}

Minor comments:
 - p. 1, item list at bottom, first item: take -> takes
 - p. 5, second paragraph: will then contained -> will then contain
 - p. 5, third paragaph: to associated -> to associate
 - Sec. 4.3, first paragraph: At the time being -> For the time being
 - below Eq. (9): most-right -> right-most
 - below Eq. (10): the second term of this equation -> the second term of the right-hand side of this equation
 - p. 6, second to last line: smaller that the -> smaller than the
 - p. 7, Sec. 5, itemize, first item: 100 millions -> 100 million
 - p. 8, last sentence: we are the -> ours is the

[Official Review · AnonReviewer3 · rating 6 · confidence 5 · 17 Dec 2016]

SYNOPSIS:
The authors introduce an efficient approximation to the softmax function that speeds up the empirical calculation of the softmax on GPUs. They leverage the unbalanced distribution of words and specific empirical timings of matrix multiplies on GPUs to devise an algorithm that selects an optimal placement of the vocabulary into clusters.  They show empirical results that show speedups over alternative methods, while not losing much accuracy compared to the full softmax. 

THOUGHTS:

Since the goal of this work is to speed up training, I'm curious why you compare only to the flat 2-level HSM (O(sqrt(V)) speedup at best), and not the deeper binary-tree HSM (O(lgV) speedup at best)?

Overall, the paper is clear, easy to understand, and well written, bar a few notation issues as pointed out by other reviewers. It adds an interesting extra tool in the language modeling toolbox. The idea is based on several previous works that aim to optimize vocabulary clustering to improve the speed-accuracy tradeoff often experienced in practice with hierarchical methods. The interesting result here seems to be that this particular clustering objective improves speed (what it was designed for), while apparently not losing much i.t.o. accuracy (what it wasn't designed for). Although the authors do not speculate  reasons for the latter part at all, I suspect it is largely related to the fact that the flat region on the timing graph (Fig 1) means that the head group V_h can actually include a sizeable portion of the most frequent words in the vocabulary at constant cost. This reduces the approximation error (regions of no support in P_approx(next | previous) compared to P_real ), which in turn mitigates the hit in perplexity compared to the full softmax. 

However, since the method is intimately related to the speed-optimal method proposed by Zweig et al. (2013) (albeit without the explicit tailoring towards GPU), I feel that a direct comparison is warranted (I understand this is underway). If the performance and accuracy improvements still hold, I will update my rating to a 7.

[Official Review · AnonReviewer4 · rating 7 · confidence 3 · 20 Dec 2016 (modified: 25 Jan 2017)]
**Final Review: nice practical speed optimization of softmax for GPUs**

The authors introduce an adaptive softmax approximation tailored for faster performance on GPUs. The key idea, which is very sensible, is to use a class-based hierarchical softmax, but where the clusters/hierarchy are distributed such that the resulting matrix multiplications are optimally-sized for GPU computation, based on their empirical tests. Their results indicate that the system does indeed work very well.

In terms of presentation, I found the paper to have both clear and unclear elements. Fortunately, the underlying concepts and logic seem quite clear. Unfortunately, at various points, the writing is not. There are various minor typos (as mentioned by AnonReviewer2, in addition to some other spots, e.g. the notation describing recurrent network in Section 3 mentions an x_t which is surely different from the x_t used in the previous paragraph on regular feedforward NN's, i think it belonged in the equation for h_t; the use of the two matrices A and P in Eq2 is strange, etc). Also, while Section 4.2 (Intuition for 2-cluster case) was a good idea to include and helpful, and while the *concepts* underlying the complexity analysis were straightforward, it could be made a lot clearer by (a) adding an additional figure such as Figure 2, along with (b) a few well-placed additional sentences unpacking the logic of the argument into easier-to-follow steps. For example, it was only when I saw Eq (6) and (7) combined with Fig(2) that the analysis on the previous page made more sense in terms of arriving at the eq for the complexity of putting the head of the distribution in the root of the tree. (Perhaps an Appendix might be the most appropriate place to add such an explanation).

[Public Comment · Edouard Grave · 21 Jan 2017]
**Response**

First and foremost, we would like to thank the reviewers for their insightful and great comments. We will edit the paper to take into account their remarks, improve its clarity and add the missing references. Second, as suggested by reviewer 2, we compared our approach to the hierarchical softmax with perplexity based clustering (referred as HSM(PPL)):

		HSM(PPL)		OURS
bg		39 (29 min)		37 (18 min)
cs		67 (55 min)		62 (30 min)
da		37 (228 min)		35 (105 min)
de		44 (207 min)		40 (110 min)
el		39 (136 min)		36 (72 min)
es		30 (194 min)		29 (103 min)

Our method obtain a slightly better perplexity, while being significantly faster. Finally, the code for our method is publicly available at:

[Final Decision · Program Chairs · 06 Feb 2017]
**ICLR committee final decision**

This is a solidly executed paper that received good reviews. However, the originality is a bit lacking. In addition, the paper would have been stronger with a comparison to the method proposed in Zweig et al. (2013). We recommend this paper for the workshop.